# Consumption of Minerals, Toxic Metals and Hydroxymethylfurfural: Analysis of Infant Foods and Formulae

**DOI:** 10.3390/toxics7020033

**Published:** 2019-06-08

**Authors:** Christian Vella, Everaldo Attard

**Affiliations:** 1Department of Pharmacy, Faculty of Medicine and Surgery, University of Malta, Msida MSD 2080, Malta; christian.g.vella@gov.mt; 2Division of Rural Sciences and Food Systems, Institute of Earth Systems, University of Malta, Msida MSD 2080, Malta

**Keywords:** infant formulae, infant foods, minerals, toxic metals, hydroxymethylfurfural, storage conditions, safety

## Abstract

Infant foods and formulae may contain toxic substances and elements which can be neo-formed contaminants or derived from raw materials or processing. The content of minerals, toxic elements, and hydroxymethylfurfural (HMF) in infant foods and formulae were evaluated. The effect of storage temperature on HMF formation in infant formulae and its potential as a quality parameter was also evaluated. Prune-based foods contained the highest HMF content. HMF significantly increased when the storage temperature was elevated to 30 °C for 21 days. All trace elements were present in adequate amounts, while the concentration of nickel was higher when compared to those of other studies. The study indicates that HMF can be used as a quality indicator for product shelf-life and that the concentrations of minerals and toxic elements vary greatly due to the diverse compositions of foods and formulae. Such contaminants need to be monitored as infants represent a vulnerable group compared to adults.

## 1. Introduction

Infants are more sensitive than adults to food contaminants due to a higher rate of uptake by the gastrointestinal tract, an incompletely developed blood–brain barrier, an undeveloped detoxification system, and high food consumption relative to body mass [1]. Heavy metals are contaminants which can accumulate in infant foods through the food chain, during food processing or leakage from packaging materials [2]. Their effect on living organisms depends on the nature and concentration of the element concerned. Some elements are an essential part of the human diet, while others can be xenobiotic and highly toxic [3]. Maximum levels for heavy metals in infant foods and formulae are only defined for cadmium, lead, and tin through Regulation (CE) No. 1881/2006 and subsequent updates [1]. Contaminants can also be formed during the heating or preservation of foods and can pose harm to human health. These are termed neo-formed contaminants. Hydroxymethylfurfural (HMF) is a neo-formed contaminant in food, being an intermediate in the Maillard reaction which consists of a series of reactions, starting with a reaction between the carbonyl group of a reducing sugar with a free amino group, or it can result from the direct dehydration of sugars [4]. It is practically not present in fresh food but it is found in variable amounts in processed foods, such as jams, fruit juices, and syrups, as its synthesis depends on the temperature, pH, concentration of saccharides, presence of organic acids, and presence of divalent ions [5].

The aim of the study was to assess the content of minerals, toxic metals (Cr, Cu, Hg, Ni, Zn, Mn and Fe), and HMF in infant foods and formulae. This would provide an insight into the potential effects of undesirable substances within a vulnerable group.

## 2. Materials and Methods

### 2.1. Sample Collection

Thirty-two infant foods from four different manufacturers were randomly selected via convenience sampling from local pharmacies and supermarkets, and categorized as apple, pear, prune, fish, poultry, and ruminant-based foods. Six infant formulae from 3 different manufacturers were randomly collected from local pharmacies and were categorized as beginner infant formulae (0–6 months) and follow-on formulae (6–12 months).

### 2.2. Determination of pH

The pH of samples was measured with a Thermo scientific Orion Star A215 pH meter (Life Technologies Ltd., Paisley, UK). For infant foods, the pH was measured directly using a probe for viscous samples while for the powdered infant formulae, a reconstitution in de-ionized water at a ratio of 1:10 was carried out.

### 2.3. Determination of HMF

HMF content was determined according to a spectrophotometric method after White [6]. The determination of HMF content was based on the determination of UV absorbance of HMF at 284 nm (SpectroStar-Nano, BMG, Labtech, Ortenberg, Germany). The difference between the absorbance of a clear sample solution and the sample solution after the addition of 0.2% NaHSO_3_ was determined to avoid the interference of other compounds at this wavelength. Five grams of each of the baby foods and infant formulae were tested for HMF content at a temperature of 18 °C. Furthermore, the infant formulae were incubated and maintained at 30 °C for 21 days in a water bath. The same HMF test procedure was used to determine the effect of temperature on HMF levels. Limits of detection (LOD) and limits of quantification (LOQ) for HMF were calculated as 3 s/m and 10 s/m, respectively, where s refers to the standard deviation of the intensity of blank samples and m refers to the slope of the calibration curve for HMF (Table 1).

### 2.4. Determination of Trace Elements

For mineral and toxic metal analysis, the samples were mineralized by digesting 1 g of the sample using 5 mL of 5% HNO_3_ at 80 °C, followed by 2 mL of 34.5–36.5% H_2_O_2_ after the acid evaporated. Further mineralization of the sample was carried out by ashing at 500 °C in a muffle furnace (Wisetherm, Wisd, Laboratory Instruments, Germany) for 6 h. The ash was reconstituted in 5 mL of 5% HNO_3_ and filtered. Deionized water was added up to 50 mL and the samples were quantitatively analyzed using a Microwave Plasma-Atomic Emission Spectrometer (MP-AES 4100, Agilent Technologies Inc., Santa Clara, CA, USA). The method was validated according to Berg [7]. The LOD and LOQ for each heavy metal were calculated as 3 s/m and 10 s/m, respectively, with respect to the calibration curve for each element (Table 1).

### 2.5. Statistical Analysis

All measurements were conducted in triplicate and average results were reported. The statistical program Prism 5 (GraphPad Software Inc., San Diego, CA, USA) was used for data analysis. The results for the heavy metal elements and hydroxymethylfurfural contents were analyzed by one-way ANOVA with the Bonferroni post hoc test to compare the statistical difference between means of the data sets and their mean difference. The same statistical test was carried out to compare the mean content of hydroxymethylfurfural between infant formulae stored at room temperature and infant formulae stored at 30 °C for 21 days. Principal component analysis and Pearson correlations were conducted on all samples, using XLSTAT v.2014.4.04 (Microsoft, version 19.4.46756, SAS Institute Inc., Marlow, Buckinghamshire, UK) to determine any clustering of minerals and toxic metals. A P value less than 0.05 was considered as statistically significant.

## 3. Results

A total of 38 samples were assessed for HMF content and selected heavy metal elements. The infant foods (*n* = 32) exhibited variable amounts of HMF, ranging from 0.89 mg/kg to 144 mg/kg, with the lowest content being present in poultry-based infant foods, while the highest content was present in prune-based products (Table 2). The HMF content in infant formulae (*n* = 6) ranged from 0.29 mg/kg to 7.87 mg/kg when examined at room temperature. The HMF content in all types of infant formulae significantly increased (*p* ≤ 0.05) after being stored at 30 °C for 21 days and ranged from 1.80 mg/kg to 9.43 mg/kg (Figure 1). The mean heavy metal content of Cr, Cu, Hg, Ni, Fe, Mn, and Zn is shown in Table 3. The trace elements were detected in all infant food and formulae samples analyzed except for Hg, which was detected only in one sample from the pear-based infant food category (*n* = 6).

## 4. Discussion

Toxic substances may be either present in the raw materials or evolve during the processing of the raw materials into the finished products. Although the assurance of food quality is the responsibility of the producer and manufacturer, authorities worldwide do not control food products for safety. Several reports have shown that baby foods may contain contaminants, some of which include microorganisms [8,9], mycotoxins [10,11], aromatic compounds [12,13], furans [14,15], and metals [16,17,18].

The HMF content was determined at a temperature of 18°C for the baby foods, and at two temperatures (18 and 30 °C) for the infant formulae. Since baby foods in individual jars are consumed within one meal and the foods have undergone extensive processing in industry, the baby foods were not tested at a temperature of 30 °C for a 21-day period. It is more likely that for infant formulae, repeated quantities are consumed from the same can over a period of time. There is no limit for the HMF content in foods, apart for honey at 40 mg/kg in general environmental conditions, 80 mg/kg for honey produced in tropical climates, and 15 mg/kg for honey with low enzymatic activity [19]. This makes it difficult to ascertain whether acceptable or excessive levels of HMF are found in the studied foods. The results from studies carried out by Kalábová and Večerek [20], and Čížková and coworkers [21], for the determination of the HMF content in infant foods, reported ranges from 2.10 mg/kg to 9.80 mg/kg and 4.10 mg to 28.90 mg/kg, respectively. The current study showed a larger spread of values nearly fifteen times the upper limit, observed by Kalábová and Večerek [20], and seven times the upper limit, observed by Čížková and coworkers [21]. This variability could be related to the type of food tested, since this varied in the different studies. A significant difference in the HMF content of prune-based infant foods compared to other infant foods (*p* ≤ 0.05) was observed and these were identified as a potential source of high HMF consumption in children. Products processed from prunes, such as pitted prunes and prune juices, have been reported to have an HMF content as high as 291 mg/kg and 528 mg/L, respectively. The higher HMF value in fruit-based foods is due to greater carbohydrate degradation as a consequence of the Maillard reaction, which is favored by a lower pH (Table 2). On the other hand, a higher furan content is present in vegetable-based foods compared to fruit-based foods. This is related to either a greater furfural content or a greater ascorbic acid degradation [14].

The HMF content in infant formulae observed in the study, ranging from 0.29 mg/kg to 7.87 mg/kg, was comparable with other studies, such as that by Michalak and coworkers [22], reporting an HMF content between 1.22 mg/kg and 8.20 mg/kg. With respect to the changes of HMF content during storage at 30 °C for 21 days, the HMF content in all formulae increased significantly after storage (*p* ≤ 0.05). This temperature-dependent effect was shown in various studies, such as that by Chávez-Servín and coworkers [23], where they demonstrated a similar significance and proportional increase after 70 days of storage at 25 °C. However, the relationship between HMF concentration and pH in infant formulae was not significant (p > 0.05). Therefore, HMF synthesis was not dependent on the pH of infant formulae. In a study conducted earlier by Chávez-Servín and coworkers [24], it was observed that infant formulae at a neutral pH for a period of 12 months of storage exhibited insignificant formation of HMF.

There was a variation in the absorbance value with respect to the concentration of the heavy metal element, and, therefore, a strong positive linear relationship was present between the two parameters (*r* = 0.9986). The low LOD and LOQ values demonstrate that the MP-AES method for the analysis of heavy metal elements was highly sensitive (Table 1). The heavy metal content varied widely due to many factors, such as differences between food types, the characteristics of the manufacturing practices and processes, and possible contamination during processing. The present study demonstrated wide variations in the concentration of the most essential and toxic elements in infant formulae and foods (Table 3). In the infant formulae, the manufacturer’s fortification of essential elements resulted in concentrations many times higher than those found in foods, especially Fe, Zn, and Cu. The concentration of nickel in the samples, ranging from 0.63 mg/kg to 1.07 mg/kg, exceeded the reference value of 5 µg/kg bw/day set by the Food and Agriculture Organization/World Health Organization (FAO/WHO) Joint Expert Committee on Food Additives (JECFA) [25], as the daily intake of Ni through infant formulae ranged from 7 μg/kg bw day to 19 μg/kg bw day. Mehrnia and Bashti [26] reported daily intake values of nickel through infant formula more than tenfold the reference value set by the JECFA. Nickel toxicity is associated with immediate and delayed hypersensitivity reactions. It has the potential to cause immunological disturbances and act as an immunotoxic agent in humans [27]. Only one sample was contaminated with Hg at a concentration of 0.7 mg/kg. Since Hg was detected in a pear-based food, the presence of methylmercury is excluded, as this bioaccumulates in fish. Therefore, this value cannot be compared to the EFSA [28], which establishes a TWI reference value of 1.3 μg/kg bw for methylmercury. Cruz and coworkers [29] reported infant formulae testing positive for mercury, with levels of 0.63 mg/kg and 0.83 mg/kg.

Factor analysis using principal components was used to identify latent traits within the data. Pearson correlation (Table 4) revealed that there were several correlations between the minerals and toxic metals. There were positive correlations between Cr and Cu, Fe, Zn (*r* = 0.718, 0.725 and 0.631), Cu and Fe, Zn (*r* = 0.996 and 0.984), and Fe with Zn (*r* = 0.974). There were negative correlations between Cu and Mn (*r* = −0.636), Mn with Fe, and Zn (*r* = −0.654 and −0.641). Two latent factors had an eigenvalue greater than 1, which together explained 80.04% of the total variance (Figure 2a). The factor loadings demonstrated the different groups of variables. For the first factor, the factor loadings of Cr, Cu, Fe, and Zn, and the second factor, weighed heavily on Hg, Ni, and Mn. Figure 2b demonstrates the factor scores of the two latent factors. Factor 1, on the horizontal axis, demonstrates the clustering of baby foods on the left hand side of the scatter plot, while the infant formulae scattered more on the right hand side. This demonstrates the distinction of the foods and formulae characteristics with respect to mineral and toxic metal values.

## 5. Conclusions

Opinions on the cytotoxicity, carcinogenic, and genotoxic potential of hydroxymethylfurfural vary, while certain minerals and toxic metals are known to be deleterious if consumed in large quantities. However, the concentrations of such metals vary depending on the food type used. Infant foods and formulae contained varying amounts of HMF and metals, thus, the total daily intake of these contaminants is affected by individual feeding patterns. Notably, a high HMF content was observed in prune-based infant foods. On the other hand, with regard to the metal contents, it was observed that infant foods contained Mn, Zn, Fe, Cu, and Cr, while infant formulae contained Zn, Fe, Cu, Mn, and Cr in decreasing order. There was a low presence of Ni and negligible quantities of Hg. Infants are within a vulnerable age group and have a restricted diet compared to other age groups, therefore, it is recommended that foods are monitored to ensure safe use. The setting up of limits with respect to this vulnerable group should be considered through further studies, using a greater diversification of samples that are subjected under varying conditions.

## Figures and Tables

**Figure 1 toxics-07-00033-f001:**
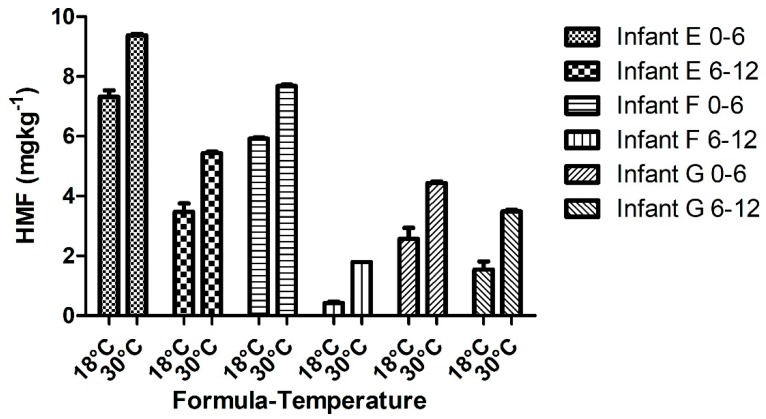
The HMF content in all types of infant formulae.

**Figure 2 toxics-07-00033-f002:**
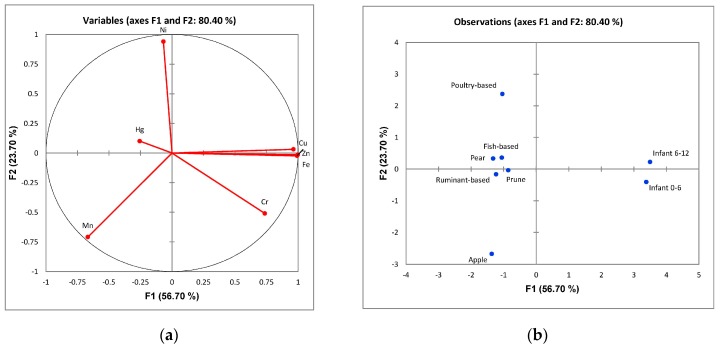
Principal component analysis (PCA) analysis of baby foods and infant formulae characteristics with respect to mineral and toxic metal contents (**a**) the factor loading plot demonstrating the different groups of variables; (**b**) the factor scores of the two latent factors.

**Table 1 toxics-07-00033-t001:** Hydroxymethylfurfural (HMF), mineral and toxic metal wavelength of detection, regression value (R^2^), limits of detection (LOD) and limits of quantification (LOQ).

Method	Element	Wavelength (nm)	*R* ^2^	LOD (mg/kg)	LOQ (mg/kg)
White	HMF	284.000	0.99000	0.1122	0.3400
MP-AES	Cr	425.433	0.99999	0.0005	0.0014
MP-AES	Cu	324.754	1.00000	0.0007	0.0022
MP-AES	Hg	253.652	0.99990	0.0789	0.2391
MP-AES	Ni	352.454	0.99998	0.0056	0.0169
MP-AES	Mn	403.076	1.00000	0.0042	0.0127
MP-AES	Fe	259.940	0.99986	0.0037	0.0113
MP-AES	Zn	213.857	1.00000	0.0301	0.0912

**Table 2 toxics-07-00033-t002:** HMF content (mg/kg) and pH of infant foods and formulae.

Mean HMF and pH Values	Prune-Based Food	Pear-Based Food	Apple-Based Food	Fish-Based Food	Poultry-Based Food	Ruminant Meat-Based Food	Formulae 0–6 Months	Formulae 6–12 Months
Mean HMF mg/kg (at 18 °C)	99.10 ± 11.45	6.327 ± 0.4945	9.674 ± 1.004	3.133 ± 0.2191	1.858 ± 0.1807	2.359 ± 0.1171	5.27 ± 1.40	1.81 ± 0.88
Mean HMF mg/kg (at 30 °C)	nd	nd	nd	nd	nd	nd	7.17 ± 1.44	3.57 ± 1.05
Mean pH	3.31 ± 0.05	3.558 ± 0.06	3.31 ± 0.04	5.64 ± 0.82	5.61± 0.16	5.17 ± 0.38	6.76 ± 0.17	6.66 ± 0.14

nd = not determined.

**Table 3 toxics-07-00033-t003:** Metal content (mg/kg) in infant foods and formulae.

Mean Metal Content (mg/kg)	Apple-Based (*n* = 6)	Pear-Based (*n* = 6)	Prune-Based (*n* = 4)	Fish-Based (*n* = 8)	Poultry-Based (*n* = 4)	Ruminant Meat-Based (*n* = 4)	Formulae 0–6 (*n* = 3)	Formulae 6–12 (*n* = 3)
Cr	0.21 ± 0.06	0.09 ± 0.03	0.18 ± 0.07	0.07 ± 0.02	0.04 ± 0.02	0.02 ± 0.01	0.29 ± 0.05	0.24 ± 0.03
Cu	0.65 ± 0.05	0.93 ± 0.11	0.66 ± 0.07	0.78 ± 0.07	0.68 ± 0.04	0.75 ± 0.07	3.33 ± 0.24	3.37 ± 0.21
Hg	nd	0.12 ± 0.12	nd	nd	nd	nd	nd	nd
Ni	0.63 ± 0.08	0.85 ± 0.03	0.86 ± 0.08	0.81 ± 0.06	1.07 ± 0.22	0.73 ± 0.06	0.76 ± 0.00	0.82 ± 0.06
Fe	0.86 ± 0.08	1.18 ± 0.26	1.67 ± 0.4	1.55 ± 0.14	1.64 ± 0.07	1.67 ± 0.41	18.34 ± 2.51	18.87 ± 3.06
Mn	4.93 ± 0.36	3.54 ± 0.06	3.22 ± 0.12	2.90 ± 0.11	2.37 ± 0.04	3.25 ± 1	2.13 ± 0.41	2.05 ± 0.21
Zn	1.07 ± 0.78	8.54 ± 8.05	1.03 ± 0.34	2.61 ± 1.46	3.19 ± 0.72	5.76 ± 0.69	27.24 ± 2.77	33.00 ± 0.95

nd = not detected.

**Table 4 toxics-07-00033-t004:** Minerals and toxic metals found in the baby foods and infant formulae.

Variables	Cu	Hg	Ni	Fe	Mn	Zn
Cr	**0.718**	−0.217	−0.393	**0.725**	−0.090	**0.631**
Cu		−0.159	−0.108	**0.996**	**−0.636**	**0.984**
Hg			0.106	−0.236	0.211	−0.062
Ni				−0.091	−0.574	−0.073
Fe					**−0.654**	**0.974**
Mn						**−0.641**

Bold values represent significant correlations.

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
