# Peer review of "Consumption of Minerals, Toxic Metals and Hydroxymethylfurfural: Analysis of Infant Foods and Formulae"

_toxics, 2019, doi:10.3390/toxics7020033_

Round 1
Reviewer 1 Report
Line 70: s and m must be defined here and not later on as they appear in the text at this point for the first time.
Line 204: The statement of this added sentence is not clear, please precise.
Author Response
Dear reviewer,
Regarding your comments:
Line 70: s and m must be defined here and not later on as they appear in the text at this point for the first time. - a reference to s and m is now included at line 70
Line 204: The statement of this added sentence is not clear, please precise. - The statements on HMF and metal contents should now be more distinctive. The first line refers to the HMF findings while the following two lines refer to the finding on metals.
We hope that our amendments are to your satisfaction.
Thanking you in advance,
Sincerely yours,
Reviewer 2 Report
The manuscript can be published without changes.
Author Response
Dear Reviewer,
Thank you for your comment.
Sincerely yours,
This manuscript is a resubmission of an earlier submission. The following is a list of the peer review reports and author responses from that submission.
Round 1
Reviewer 1 Report
I include here some comments to take into account before the final decision related to the publication of this manuscript:
The term “minerals” used in the title and throughout the manuscript should be replaced by inorganic elements or metals.
There is no data related to the validation of the analytical methods used: the spectrophotometric method employed for the determination of HMF has been developed for honey and not for baby foods. Moreover, MP-AES is a plasma with less energy than the ICP and the influence of the matrix and other ionisable elements in the samples should be investigated.
The text included in the conclusions section does not reflect the main conclusions of the work. Please modify it accordingly.
Reviewer 2 Report
The authors are here presenting a manuscript focused on the determination of 5-hydroxymethyl-2-furaldehyde and some selected metals in infant foods and formulae. The manuscript is well written and correctly structured.
Despite the relevance of the topic and the generally correct analytical approach used to perform the experiments, unfortunately my opinion is that the work lacks some originality.
In fact, scientific literature already offers several examples of studies devoted to quantify the HMF content and heavy metals in this type of samples. And also the effect of temperature and time on furfural related compounds levels has already been investigated.
My suggestion is that the author conduct the studies on a definitely larger number of samples (so that also the chemometric treament could acquire a higher reliability), for example focusing on a specific market in order to enhance the originality aspects of the work.
Reviewer 3 Report
The originality of the study and the novelty it brings in the field is not very clear- try to highlight them; A creative step furthering the knowledge in the field must be evident.
Abstract clearly presents objects, methods and results
Line 58- close the parenthesis
Conclusions are too general and not derived from the data presented
References - check carefully, use abbreviation such as Plant Sci. Today., etc. – not full names of journals, e.g. Food Control, etc.
Reviewer 4 Report
The manuscript is well written but there are still some questions:
What are the LOD and LOQ of the HMF method?
Is the AES-method for the trace metals a validated method? If not repeatability and recovery should be added. Otherwise add a reference.
It is not clear why just these trace metals have been chosen for analysis. Why cadmium and lead for which maximum levels are defined have not been considered?
Are there other amounts for trace metals in infant food and formulae known from the literature? Please compare.